# Comprehensive Transcriptomic Profiling of m6A Modification in Age-Related Hearing Loss

**DOI:** 10.3390/biom13101537

**Published:** 2023-10-18

**Authors:** Menglong Feng, Xiaoqing Zhou, Yaqin Hu, Juhong Zhang, Ting Yang, Zhiji Chen, Wei Yuan

**Affiliations:** 1Chongqing Medical University, Chongqing 400016, China; 2Chongqing Institute of Green and Intelligent Technology, Chinese Academy of Sciences, Chongqing 400714, China; 3Chongqing School, University of Chinese Academy of Sciences, Chongqing 400714, China; 4Department of Otolaryngology & Head and Neck, Chongqing General Hospital, Chongqing 401147, China

**Keywords:** age-related hearing loss, m6A modification, MeRIP-Seq, RNA-Seq

## Abstract

Age-related hearing loss (ARHL), also known as presbycusis, is one of the most common neurodegenerative disorders in elderly individuals and has a prevalence of approximately 70–80% among individuals aged 65 and older. As ARHL is an intricate and multifactorial disease, the exact pathogenesis of ARHL is not fully understood. There is evidence that transcriptional dysregulation mediated by epigenetic modifications is widespread in ARHL. However, the potential role of N6-methyladenosine (m6A) modification, as a crucial component of epigenetics, in ARHL progression remains unclear. In this study, we confirmed that the downregulation of m6A modification in cochlear tissues is related to ARHL and found that the expression of the m6A methylation regulators Wilms tumour suppressor-1-associated protein (WTAP), methyltransferase-like 3 (METTL3), ALKB homologous protein 5 (ALKBH5) and fat mass and obesity-associated protein (FTO) is decreased significantly at the mRNA and protein levels in ARHL mice. Then, we used methylated RNA immunoprecipitation sequencing (MeRIP-Seq) and RNA sequencing (RNA-Seq) to identify the differentially m6A-methylated genes in the cochlear tissues of ARHL mice. A total of 3438 genes with differential m6A methylation were identified, of which 1332 genes were m6A-hypermethylated and 2106 genes were m6A-hypomethylated in the ARHL group compared to the control group according to MeRIP-seq. Further joint analysis of RNA-Seq and MeRIP-Seq data showed that 262 genes had significant differences in both mRNA expression and m6A methylation. GO and KEGG analyses indicated that 262 unique genes were enriched mainly in the PI3K-AKT signalling pathway. In conclusion, the results of this study reveal differential m6A methylation patterns in the cochlear tissues of ARHL mice, providing a theoretical basis for further study of the pathogenesis of ARHL and potential therapeutic strategies.

## 1. Introduction

Age-related hearing loss (ARHL), one of the three most common chronic diseases among older people, has a prevalence of approximately 70–80% among individuals aged 65 and older and has gradually become a major global public health issue [1]. ARHL is caused by age-related degeneration of the cochlear and the central auditory system, and its main clinical manifestations are high-frequency hearing loss and decreased speech comprehension (especially in noisy environments) [2,3]. As the main cause of sensory disorders, ARHL limits the ability of elderly individuals to communicate with their families and society, leading to social isolation, anxiety, depression, and cognitive decline [4]. Although hearing aids and cochlear implant surgery have been shown to significantly improve hearing loss in patients with ARHL, the majority of patients are still not effectively treated due to the inconvenience of treatment or high medical expenses [5]. Hair cells and spiral ganglion neurons are the two main cell types responsible for hearing in the cochlea [6]. Current research has found that a reduction in cell numbers due to irreversible degeneration and death of inner hair cells and spiral ganglion neurons in the cochlea is the primary pathological change in ARHL [7]. In addition, oxidative stress, mitochondrial DNA deletions or mutations, reduced autophagy, disorders of ion transport across the membrane and damage to vascular endothelial cells have been found to be closely involved in the above pathological changes [8]. Nevertheless, the precise molecular mechanisms and signalling pathways underlying ARHL have not yet been fully elucidated.

RNA modifications, as the crucial components of epigenetics, have been found to play important regulatory roles in various physiological processes and disease progression [9]. N6-methyladenosine (m6A) methylation is widely acknowledged as the most common modification observed in eukaryotic mRNAs and exerts crucial influences on various aspects of mRNA biology, including RNA splicing, transport, translation, and decay [10]. The process of m6A methylation is characterised by a dynamic nature and reversibility and is regulated by three types of enzymes known as “writers”, “erasers”, and “readers” [11]. Writers are m6A methyltransferases, mainly including Wilms tumour suppressor-1-associated protein (WTAP), methyltransferase-like 14 (METTL14), and methyltransferase-like 3 (METTL3). Erasers, known as m6A demethylases, include only two enzymes: ALKB homologous protein 5 (ALKBH5) and fat mass and obesity-associated protein (FTO). Additionally, readers play an essential role in the biological function of m6A methylation by recognizing and binding m6A methylation sites to regulate mRNA metabolism. The main readers are insulin-like growth factor 2 mRNA binding proteins (IGF2BPs) and the YT521-B homology (YTH) domain family (YTHDC1/2 and YTHDF1–3) [12].

Increasing evidence indicates that m6A methylation and its regulators are related to multiple ageing-associated diseases. Guo et al. showed that METTL3 positively regulates *JAK2* mRNA expression levels to promote angiogenesis and atherosclerosis by increasing m6A methylation of *JAK2* mRNA, which is facilitated by the recognition and binding of IGF2BP1 to the m6A site in *JAK2* mRNA [13]. Chen et al. discovered a decrease in the global m6A modification of mRNAs in PC12 cells induced by 6-OHDA and in the striatum of the Parkinson’s disease rat brain. Further research confirmed that the decrease in m6A level promoted the expression of N-methyl-d-aspartate receptor 1 and alleviated Ca^2+^ influx and oxidative stress, leading to dopaminergic neuron apoptosis [14]. Shafik et al. revealed that METTL3 is downregulated in the hippocampus in Alzheimer’s disease (AD) patients and that the levels of METTL3 and m6A modification of AD-related mRNAs are also reduced in 5XFAD mice [15], while upregulation of METTL3 can promote *STUB1* gene expression through an m6A-IGF2BP1-dependent mechanism involving stabilisation of *STUB1* mRNA and promotion of autophagic p-Tau clearance in Aβ1-42-treated cells, thus improving AD [16]. Moreover, a recent study found that there are decreased global m6A modification and METTL3 levels in the blood of ARHL patients and the cochleae of mice. Further studies have revealed that METTL3 reduces apoptosis in ARHL by regulating the m6A level of *SIRT1* mRNA in an IGF2BP3-dependent manner [17]. However, m6A methylome profiles in ARHL have never been reported.

In this study, we mapped the trend in m6A level in the mouse cochlea with age and quantified the expression of *METTL3*, *WTAP*, *FTO*, and *ALKBH5* genes in the cochleae of ARHL mice compared with control mice. In addition, we acquired, for the first time, the transcriptome-wide m6A profile of ARHL mice by using a combination of methylated RNA immunoprecipitation sequencing (MeRIP-Seq) and high-throughput RNA sequencing (RNA-Seq) and found considerable numbers of significantly differentially expressed m6A-modified transcripts and related pathways. Our study reveals the important roles of m6A-modified transcripts in the molecular mechanisms of ARHL and identifies potential therapeutic targets of ARHL.

## 2. Materials and Methods

### 2.1. Animals

All experiments were conducted in C57BL/6J male mice, and a total of 96 male C57BL/6J mice were included in our experiments. The mice were divided into five groups: six-week-old (6 w) (*n* = 39), 3-month-old (3 m) (*n* = 6), 6-month-old (6 m) (*n* = 6), 9-month-old (9 m) (*n* = 6) and 12-month-old (12 m) (*n* = 39) mice. Of these, 6 each of the 6 w, 3 m, 6 m, 9 m and 12 m mice were used for m6A quantification; 9 each of the 6 w and 12 m mice were used for quantitative real-time PCR (qRT–PCR) and western blotting; 15 each of the 6 w and 12 m mice were used for MeRIP-Seq and RNA-Seq; and 9 each of the 6 w and 12 m mice were used for MeRIP-qPCR. All procedures involving animal use and care were approved by the Chongqing Medical University Animal Welfare Committee.

### 2.2. Auditory Brainstem Response (ABR) Analysis

The mice were anaesthetised using 2% isoflurane for induction and 1.5% isoflurane for maintenance before each measurement. The mouse body temperature was maintained at 37–38 °C by keeping the anaesthetised mice on a heating pad. All tests were performed in a soundproof chamber. The acoustic signals of click and 4-, 8-, 16-, 24- or 32 kHz pure tone bursts were channelled into the mouse ear canals. The amplified brainstem responses were recorded from the scalps of the mice by Intelligent Hearing Systems ( Smart-EP software v3.30, Miami, FL, USA) through subdermal electrode needles inserted on the overlying and ventral vertex regions of the left and right bullae. The stimulus intensities were recorded in 5-dB SPL intervals down from a maximum stimulus intensity to identify the minimum stimulation level at which no waveform could be visualised, and the minimum stimulation level was determined as the ABR threshold.

### 2.3. RNA Extraction and qRT–PCR

Total RNA was extracted from mouse cochlear tissue using TRIzol Reagent (Thermo Fisher Scientific, Waltham, MA, USA) according to the manufacturer’s instructions. DNase I was used to digest DNA after RNA extraction. A260/A280 ratios were calculated to detect the quality of RNA using a NanoDrop^®^ ND-1000. PrimeScript™ RT Master Mix (RR036A, Takara Biotechnology Co., Ltd., Dalian, China) was used to convert total RNA to cDNA following the manufacturer’s protocol. qRT–PCR was performed using a TB Green^®^ Premix Ex Taq™ II (RR820A, Takara Biotechnology Co., Ltd., Dalian, China) and LightCycler 480 system (LightCycler480 software 1.5, Roche, Basel, Switzerland). The conditions of the qRT–PCR cycle included a predenaturation step at 95 °C for 30 s followed by 40 cycles of 95 °C for 5 s and 60 °C for 20 s. The mRNA primers used in our study are listed in Table A1.

### 2.4. m6A Quantification

The alterations in global m6A levels in total mRNA were detected with an EpiQuik m6A RNA Methylation Quantification Kit (Colorimetric) (Epigentek, Farmingdale, NY, USA) following the manufacturer’s protocol. Two hundred nanograms of poly-A-purified RNA was used for sample analysis. Signals of m6A were detected at 450 nm.

### 2.5. Western Blot Analysis

Total protein was extracted using RIPA lysis buffer containing 1% phosphatase inhibitors and 1% PMSF and quantified using a BCA protein assay kit (Beyotime, Shanghai, China). Protein samples (25 μg) were separated with 4–12% SDS–PAGE and transferred from the gel to PVDF membranes (Merck Millipore, Kenilworth, NJ, USA). Then, the PVDF membranes were blocked with QuickBlock™ western solution (Beyotime, Shanghai, China) for 10 min and incubated with METTL3, WTAP, ALKBH5, FTO, and GAPDH primary antibodies at 4 °C overnight (dilution 1:2000; Sanying, Wuhan, China). The membranes were subsequently incubated with the secondary antibody, goat anti-rabbit IgG (dilution 1:4000; Sanying, Wuhan, China), for 2 h at room temperature. SuperSignal™ West Femto Maximum Sensitivity Substrate (Thermo Fisher Scientific, Waltham, MA, USA) was used to develop the bands. ImageJ software (ImageJ software v1.6.0, NIH, Bethesda, MD, USA) was used to quantify western blot results.

### 2.6. MeRIP-Seq and Data Analysis

MeRIP-Seq was completed by Shanghai Cloud-Seq Biotech, and m6A immunoprecipitation was performed by using a m6A RNA Methylation Kit (GenSeq Inc., Shanghai, China). Fifteen mice were randomly selected from each of the 6 w and 12 m groups, and each group of mice was further randomly divided into three mixed biological samples (five in each sample). TRIzol reagent (Thermo Fisher Scientific, Waltham, MA, USA) was used to extract total RNA from each sample according to the manufacturer’s instructions. A NanoDrop^®^ ND-1000 and agarose gel electrophoresis were used to detect the quantity, integrity, and quality of total RNA from the above six samples. The RNA was fragmented into approximately 200-nt fragments by RNA Fragmentation Reagents (GenSeq Inc., Shanghai, China) as instructed by the manufacturer and reserved 1 μg of fragmented RNA as the input control. Briefly, the tubes with 1 μg/μL RNA and fragmentation buffer (metal-ion) were incubated at 70 °C for 6 min in a preheated thermal cycler block. Subsequently, 30 μL of 3M sodium acetate (NaOAc),1 μL of glycoBlue, and 750 μL volumes of 100% ethanol were added, mixed, and kept at −80 °C overnight. After centrifugation and pellet resuspension. Then, fragmented RNA was incubated with 5 μg of anti-m6A polyclonal antibody (Cat. No. 202003, Synaptic Systems, Goettingen, Germany) in IPP buffer for 2 h at 4 °C. The mixture was then immunoprecipitated by incubation with protein-A beads (Thermo Fisher Scientific, Waltham, MA, USA) at 4 °C for an additional 2 h. The RNA was eluted from the RNA-binding protein complexes with m6A in IPP buffer, and then the RNA was extracted with TRIzol reagent (Thermo Fisher Scientific, Waltham, MA, USA) following the manufacturer’s instructions. Purified RNA was used for RNA-seq library generation with the GenSeq^®^ Low Input Whole RNA Library Prep Kit (GenSeq Inc., Shanghai, China) and an Agilent 2100 Bioanalyzer (Agilent Technologies, Inc., Santa Clara, CA, USA) was used to assess the quality of the libraries. Both the input sample without immunoprecipitation and the m6A IP samples were subjected to 150 bp paired-end sequencing on an Illumina HiSeq 6000 sequencer (Illumina, San Diego, CA, USA).

After sequencing, the Q30 of the raw data was determined for assessment of sequencing quality. Cutadapt (v1.9.3) (https://journal.embnet.org/index.php/embnetjournal/article/view/200, accessed on 6 October 2023) software was used to remove adaptor sequences and low-quality reads. The clean reads of all samples were matched to the mouse reference genome (mm10) using HISAT2 (v2.0.4) software. MACS (v1.4.2) (http://liulab.dfci.harvard.edu/MACS/ accessed on 6 October 2023) software was used to identify methylated genes in each sample under the parameters: “(-g): 1.27 × 10^9^, *p*-value < 0.00001”. Next, DREME (v5.3.0) (DREME Tutorial - MEME Suite (http://memesuite.org/, accessed on 6 October 2023) software was utilised to analyse the motifs of the m6A peaks with the criteria of E-value < 1.0 × 10^−10^. The diffReps (v1.55.6) (https://code.google.com/p/diffreps/under, accessed on 6 October 2023) software identified differentially methylated genes between 6 w mice and 12 m mice with the criteria of a *p*-value < 0.00001 and a fold change ≥ 2. Kyoto Encyclopedia of Genes and Genomes (KEGG) and Gene Ontology (GO) analyses were performed on differentially methylated genes by using the KEGG (www.genome.jp/kegg, accessed on 6 October 2023) and GO (www.geneontology.org, accessed on 6 October 2023) databases.

### 2.7. RNA-Seq and Data Analysis

The same total RNA samples employed for MeRIP-Seq were used for RNA-Seq. RNA-Seq was also completed by Shanghai Cloud-Seq Biotech. A GenSeq^®^ rRNA Removal Kit (GenSeq Inc., Shanghai, China) was used to remove the ribosomal RNA from RNA samples. After the removal of rRNA, the above samples were applied to construct an RNA sequencing library following the manufacturer’s instructions with a GenSeq^®^ Low Input RNA Library Prep Kit (GenSeq Inc., Shanghai, China). Quality control and quantitation of the constructed RNA sequencing library were carried out with a Bioanalyzer 2100 system (Agilent Technologies, Inc., Santa Clara, CA, USA), and the libraries were subsequently sequenced (150 bp, paired-end reads) on an Illumina NovaSeq 6000 instrument (Illumina, San Diego, CA, USA).

After sequencing, the Q30 of the raw data was determined to assess sequencing quality. Cutadapt (v1.9.3) (https://journal.embnet.org/index.php/embnetjournal/article/view/200, accessed on 6 October 2023) software was used to remove adaptor sequences and low-quality reads. The clean reads of all samples were matched to the reference genome using HISAT2 (v2.0.4) (https://daehwankimlab.github.io/hisat2/, accessed on 6 October 2023) software. HTSeq (v0.9.1) (High-throughput sequence analysis in Python-HTSeq 2.0.4 documentation) software was used to calculate the raw counts. Then, edgeR was used for standardisation, and the fold change and *p*-value between the two groups of samples were calculated to screen for differentially expressed mRNAs with the criteria of a *p*-value < 0.05 and a fold change ≥ 2. KEGG and GO analyses of differentially expressed mRNAs were performed as described above.

### 2.8. MeRIP-qPCR

MeRIP-qPCR was performed by Shanghai Cloud-Seq Biotech. This experiment consisted of two parts: MeRIP and qRT–PCR. The experimental flow of MeRIP and qRT–PCR was performed as described previously. Briefly, the fragmented RNA was subjected to immunoprecipitation with 5 μg of anti-m6A polyclonal antibody (Cat. No. 202003, Synaptic Systems, Goettingen, Germany) coated magnetic beads. Methylated RNA bound to the magnetic beads was eluted and purified. Finally, the m6A modification was verified by qPCR analysis. The sequences of the primers used in MeRIP-qPCR are listed in Table A2. Percentage (IP/Input) was used to calculate the modification difference between the two groups.

## 3. Results

### 3.1. Increased Hearing Thresholds and Decreased Cochlear m6A Modification in Ageing Mice

The C57BL/6J mouse strain is known for early-onset hearing loss and is one of the most extensively used experimental models for studying ARHL. To explore the change in hearing with ageing, the hearing thresholds of click, 4 kHz, 8 kHz, 16 kHz and 32 kHz in 6-week-old (6 w), 3-month-old (3 m), 6-month-old (6 m), 9-month-old (9 m) and 12-month-old (12 m) C57BL/6J mice were measured by auditory brainstem response (ABR) assessment. We found that the 6 w mice showed normal mean hearing thresholds (<30 dB SPL), whereas the 12 m mice displayed severe hearing loss (>80 dB SPL) (Figure 1a–e). In addition, the hearing thresholds of mice showed a significantly upwards shift with ageing at all frequencies, and the hearing thresholds of 12 m mice were highest at all frequencies (Figure 1a–e). Previous reports have shown that C57BL/6J mice exhibit the classic pattern of ARHL by 12 to 15 months of age [18], which is similar to our above results. Therefore, 12-month-old C57BL/6J mice were selected as the experimental model of ARHL, and 6-week-old C57BL/6J mice were selected as the normal controls (NCs) in this study.

Moreover, to investigate whether ARHL is related to the alteration of m6A modification, the m6A levels in the cochleae of 6 w, 3 m, 6 m, 9 m, and 12 m mice were compared by colorimetric m6A quantification assay. As indicated in Figure 1f, the m6A levels in the cochleae of mice showed overall downward trends with ageing, and the total m6A levels in 9 m and 12 m mice were significantly lower than those in 6 w mice. These results suggest that the development of ARHL is accompanied by changes in the level of m6A in the cochlea.

### 3.2. Changes in m6A-Related Modification Enzymes in the Cochleae of 6 w and 12 m Mice

To explore the changes in m6A-related modification enzymes in ARHL, we used qRT–PCR and western blotting to measure the expression of genes associated with methylases and demethylases in the cochleae of 6 w and 12 m mice, including *WTAP*, *METTL3*, *FTO*, and *ALKBH5* genes. As shown in Figure 2a–d, we found that the expression of *WTAP*, *METTL3*, *FTO*, and *ALKBH5* genes was significantly downregulated in the 12 m mice compared with the 6 w mice at the mRNA level. In addition, we further measured the protein expression of the above four enzymes (Figure 3a–d). The levels of WTAP, METTL3, FTO, and ALKBH5 were also significantly downregulated in 12 m mice compared to 6 w mice, which was generally consistent with the pattern of changes at the mRNA level. Therefore, the combined effect of the downregulation of WTAP, METTL3, FTO, and ALKBH5 might be responsible for the decrease in total m6A in the cochlear RNA of ARHL mice.

### 3.3. General Characteristics of m6A Methylation Modification in ARHL

MeRIP-Seq was conducted to elucidate the characteristics of m6A methylation in 12 m mouse and 6 w mouse cochlear tissues. The bar graph shows the levels of m6A methylation in different groups. We found an average of 30,957 m6A peaks in 6-week-old mice and 28,147 m6A peaks in 12-month-old mice (Figure 4a). Subsequently, we mapped the above m6A peaks to 11,633 annotated genes in 6 w mouse cochlea tissues and 10,963 annotated genes in 12 m mouse cochlea tissues, of which 10,566 annotated genes were common between the two groups (Figure 4b). All m6A-methylated genes were split into groups based on the number of peaks in each gene (Figure 4c). Of note, most genes had two or more m6A peaks in the 6 w mice and 12 m mice. However, the number of peaks per gene in the 12 m mice was lower than that in the 6 w mice, which also demonstrated that the m6A modification level decreased after ARHL. In addition, our results revealed that the distribution of m6A peaks was different between the 6 w mice and 12 m mice, but both were enriched mainly in coding sequences (CDSs), stop codons (stopC), and start codons (startC) of mRNA (Figure 4d,e). More specifically, there were relative decreases in CDSs (6 w: 52.9%, 12 m: 47.5%) and stopC (6 w: 23.8%, 12 m: 16.5%), and relative increases in the numbers of m6A peaks in startC (6 w: 15.0%, 12 m: 16.7%), 3′ untranslated regions (3′UTR) (6 w: 1.9%, 12 m: 6.9%), and 5′ untranslated regions (5′UTR) (6 w: 6.4%, 12 m: 12.4%) (Figure 4e).

Considering that the binding of various binding proteins to specific motifs is the premise for the onset of RNA methylation and demethylation, we conducted motif enrichment analysis on the m6A peaks. As shown in Figure 4f, the top-ranking motif structure was characterised as DGAAGH (D = A/G/U; H = A/C/U) in both groups. Analysis of the m6A methylation enrichment in different chromosome loci showed that the m6A methylation peaks of chromosomes were increased on the X and Y chromosomes of the 12 m mice compared with the control mice but decreased in the remaining chromosomes (Figure 4g,h). Furthermore, in the 6 w and 12 m mice, the chromosomes with the highest m6A methylation frequency were chromosome 2, chromosome 11, and chromosome 1, with 3378 and 3268, 3173 and 3014, and 2997 and 2826 m6A methylation peaks, respectively (Figure 4h). However, further comparison showed that there was no significant difference in the enrichment number of m6A methylation peaks on chromosomes between the 12 m and 6 w groups.

### 3.4. Functional Enrichment and Pathway Analysis of Differentially m6A-Methylated Genes

We generated volcano plots and heatmaps to visually display the significant differentially methylated genes (12 m vs. 6 w). As shown in Figure 5a,b, a total of 3438 genes with differential m6A methylation were identified, of which 1332 genes were m6A-hypermethylated and 2106 genes were m6A-hypomethylated in the ARHL group compared to the control group (*p*-value < 0.00001 and |fold change| ≥ 2).

Furthermore, GO analysis was performed to gain further insights into the potential functions of the differentially methylated genes in 12 m mice. For the biological process (BP) category (Figure 6a,e), the genes with hypermethylated or hypomethylated m6A-modified peaks were all significantly enriched with metabolism-related terms, such as metabolic process, cellular metabolic process, and primary metabolic process. For the cellular component (CC) category (Figure 6b,f), the genes with hypermethylated or hypomethylated m6A-modified peaks were all significantly enriched in intracellular anatomical structure, organelle, intracellular organelle, etc. For the molecular function (MF) category (Figure 6c,g), the differentially methylated genes were associated with binding, protein binding, ion binding, etc.

KEGG analysis was further performed to explore the potential pathways associated with the differentially methylated genes. The genes with hypermethylated m6A-modified peaks were significantly enriched in several key pathways, including RNA degradation and the thyroid hormone signalling pathway (Figure 6d). However, the genes with hypomethylated m6A-modified peaks were significantly associated with several key pathways, including the MAPK signalling pathway, Ras signalling pathway, and Rap1 signalling pathway (Figure 6h).

### 3.5. Combined Analysis of MeRIP-Seq and RNA-Seq Data

The transcriptional profiles of the 6 w and 12 m mice were obtained from RNA-Seq. We found 1241 differentially expressed genes in the ARHL group compared with the control group, 290 upregulated and 951 downregulated (*p*-value < 0.05 and |fold change| ≥ 2) (Figure 7a,b). Further combined analysis of differentially expressed genes and differentially methylated m6A peaks identified a total of 262 genes that were significantly differentially expressed (Figure 7c). Of those, compared with the control group, 27 hypermethylated m6A peaks were distributed in 19 upregulated genes, 182 hypermethylated m6A peaks in 67 downregulated genes, 42 hypomethylated m6A peaks in 25 upregulated genes, and 276 hypomethylated m6A peaks in 137 downregulated genes (Figure 7c,d). Furthermore, there were both hypomethylated m6A peaks and hypomethylated m6A peaks in 1 upregulated gene and 13 downregulated genes (Figure 7c). To illustrate the role of m6A modification in ARHL, 262 genes with both differentially methylated m6A peaks and differential expression were selected for further GO and KEGG analyses. The top highly enriched GO terms are shown in Figure 7e–g. These included cellular metabolic process, metabolic process, regulation of cellular metabolic process (BP category), binding, protein binding, enzyme binding (MF category), and intracellular anatomical structure, organelle, and intracellular organelle (CC category). In addition, as shown in Figure 7h, the PI3K-AKT signalling pathway was highly enriched among the KEGG pathways of those genes.

### 3.6. Validation of Differentially m6A-Modified Genes by MeRIP-qPCR and qRT–PCR

To verify the reliability of the MeRIP-Seq and RNA-Seq results, MeRIP-qPCR and qRT–PCR were used to verify the modification levels of modification sites on the target genes and the gene expression levels, respectively. We selected the top two hypermethylated genes and the top two hypomethylated genes with the most significant differences in methylation levels as target genes from among 262 genes with significant differences in both methylated m6A peaks and mRNA expression levels. The MeRIP-qPCR and qRT–PCR results of *RAPGEF6*, *BIRC6*, *RPS6KA3*, and *SH2D1B1* genes were in line with the sequencing results, and there were significant differences (Figure 8a–h). These results suggested that our sequencing results were accurate and credible. The information of the top 10 hypermethylated genes and top 10 hypomethylated genes among the 262 genes with differentially expressed mRNA and differentially methylated m6A peaks were listed in Table 1.

## 4. Discussion

ARHL, one of the most common sensory disorders, has become the main cause of communication deficits in elderly individuals and seriously affects their quality of life [19]. A growing number of studies have revealed that epigenetic mechanisms, including DNA methylation, histone modification, and noncoding RNA activities, are involved in the development of ARHL [20,21]. For example, Xu et al. have reported that DNA hypermethylation of the *SLC26A4* gene results in an increased risk for ARHL in men [22], while DNA hypermethylation of the *KCNQ5*, *ERBB3*, and *SOCS3* genes results in an increased risk for ARHL in women [23]. Wang et al. found that acetylated histone H3 can be detected in spiral ganglion cells and the organ of Corti in young mouse cochleae, while it is absent in aged mouse cochleae [20]. Conversely, dimethylated histone H3 can be detected in aged mice but not in young mice. In addition, Chen et al. indicated that upregulation of miRNA expression in pathways promoting apoptosis and inhibiting autophagy and downregulation of miRNA expression in pathways promoting proliferation and differentiation are involved in the progression of age-related hearing loss [24]. In recent years, one of the crucial components of epigenetics, m6A methylation, has been reported to be associated with a variety of ageing-associated degenerative diseases, such as chronic obstructive pulmonary disease, atherosclerosis, and age-related macular degeneration [13,25,26]. However, there are limited studies regarding m6A methylation in the pathological progression of ARHL.

To investigate whether m6A modification plays a role in the development of ARHL, we first investigated the association between m6A levels and ARHL. We found that ABR thresholds of experimental model mice increased with increasing age, but total RNA m6A modification in the cochleae of experimental model mice declined gradually. The level of m6A modification in the cochleae of 12 m mice was the lowest, and the ABR thresholds were greatest in these mice. In addition, Wang et al. reported similar findings that the levels of m6A modification in the total RNA of blood cells in ARHL normal patients were significantly lower than those in elderly patients without hearing loss [17]. These results suggest that the downregulation of m6A modification in the cochlea may be correlated with the development of ARHL. Considering that m6A modification is directly regulated by methyltransferases and demethylases [11], we measured the expression of genes encoding four common regulators in the cochleae of 12 m and 6 w mice, including *WTAP*, *METTL3*, *FTO*, and *ALKBH5*. Interestingly, at both the mRNA and protein levels, the expression of the methyltransferases *WTAP* and *METTL3* genes was decreased in 12 m mice, and the expression of the demethylases *FTO* and *ALKBH5* genes was also decreased significantly. This suggests that there may be a simultaneous occurrence of hypermethylation and hypomethylation of numerous genes in the cochlear tissues of ARHL mice and that the overall decrease in m6A level may be a result of coordinated regulation by multiple m6A enzymes.

To better clarify the changes in methylation modification and gene expression in cochlear tissues during ARHL, we constructed MeRIP-Seq and RNA-Seq libraries for high-throughput sequencing and performed a series of bioinformatics analyses on the results. We found that the transcripts of 11,633 genes had m6A modifications in 6 w mice and the transcripts of 10,963 genes in 12 m mice. In further analysis, 3438 differentially methylated genes between 6 w and 12 m mice, of which 1332 m6A hypermethylated genes and 2106 m6A hypomethylated genes were identified in 12 m mice. These results were consistent with our previous finding of a significant decrease in overall methylation level in 12 m mice, suggesting that our MeRIP-Seq results were reliable. Subsequent analysis revealed that although the degree of m6A methylation of genes was different between the two groups, the distribution and pattern of m6A methylation were similar, with major enrichment observed in the CDS region. Nevertheless, m6A deposition in the CDS region has been reported to tend to stabilize mRNA and be dynamic and reversible [27,28], which suggests that m6A modification may have a major role in regulating the expression of genes related to ARHL. Previous studies have revealed that the RRACH motif (R = A/G; H = A/C/U), the core sequence of m6A modification sites, is critical for m6A methylation [29]. The common m6A regulatory molecules WTAP, METTL14, and METTL3 have been shown to interact with the RNA motifs GACU, GGAC, and GGAC, respectively [30]. However, Gilbert et al. showed that not all RRACH sites in the body undergo m6A modification [31], which is consistent with our findings that the top-ranking motif structure in both groups was DGAAGH (D = A/G/U; H = A/C/U). This suggests that m6A modification in ARHL may be regulated by other molecular mechanisms and that further studies are needed. In addition, we also found many m6A motifs in 6 w and 12 m mouse tissues with some differences in sequence, further suggesting the possible presence of specific m6A methylation sites in ARHL.

In our study, GO and KEGG analyses of differentially m6A-modified genes were performed to explore their functions. Interestingly, in the GO analysis, we found that genes with significant hypomethylation or hypermethylation were mostly enriched with the same BP, MF, and CC terms, which suggests that these differentially methylated genes may have the same subcellular localisation and molecular biological functions. Of those, genes with significant hypomethylation or hypermethylation were mostly enriched in metabolic biological processes. Coling et al. reported that *SOD1* gene deficiency increases the vulnerability of the cochlea to damage associated with ageing and noise through metabolic pathways involving the superoxide radical [32]. In addition, the overactivation of AMP-activated protein kinase leads to accelerated ARHL in Tg-mtTFB1 mice through redox imbalance and dysregulation of the apoptotic pathway, whereas a reduction in AMP-activated protein kinase can suppress cochlear cell apoptosis through the ROS-AMPK-BCL2 pathway [33]. These results are consistent with those of our GO analysis showing that metabolic enzymes and related regulatory factors may be involved in the occurrence of ARHL. Unlike the results obtained for GO analysis, the results of KEGG pathway analysis of hypomethylated genes were significantly different from those of hypermethylated genes. Among them, the genes with hypomethylated m6A-modified peaks were mainly associated with the MAPK signalling pathway. Previous evidence has shown that the MAPK signalling pathway plays an important role in regulating cell proliferation, apoptosis, autophagy, and differentiation [34], and the activated MAPK signalling pathway can promote the senescence of various cells and tissues [35,36]. However, inhibition of the MAPK signalling pathway has been confirmed to alleviate noise or drug-induced hair cell apoptosis and hearing loss [37,38], suggesting that ARHL might also be related to alteration of the MAPK signalling pathway. In addition, our results revealed that the genes with hypermethylated m6A-modified peaks were mainly enriched in RNA degradation. It is well known that RNA degradation is a major component of overall RNA metabolism and plays an important role in regulating levels of RNA expression [39]. However, RNA degradation is also an important way for m6A modification to exert biological functions [40], indicating that some genes with hypermethylated m6A-modified peaks might participate in the development of ARHL by altering RNA levels.

According to the joint analysis of MeRIP-Seq and RNA-Seq data, we found 262 genes with differential expression and differential m6A methylation simultaneously. Subsequently, the above differentially expressed genes were analysed via GO and KEGG analyses, and the main enriched biological processes were metabolic processes, while the main enriched pathway was the PI3K-AKT signalling pathway. Notably, in our study, most of the genes enriched in the PI3K-AKT signalling pathway are positive regulators of this pathway, including *COL1A2*, *GHR*, *IL7R*, *ITGA6*, *LPAR1*, and *THBS4* genes. However, the expression of these genes was significantly downregulated in 12 m mice compared to the control group (Table A3). The PI3K-AKT signalling pathway serves an important role in regulating cell proliferation, differentiation, apoptosis, and senescence [41]. Previous studies have found that inhibition of the PI3K-AKT signalling pathway can induce cell senescence, and directly increase the expression of p21 [42,43]. Additionally, activation of this pathway has also been found to reduce hair cell loss and hearing damage in cisplatin-induced and noise-induced hearing loss [44,45]. These results suggested that the alteration of the PI3K-AKT signalling pathway may also play an important role in ARHL. In the following, compared with the control group, we further found a total of 86 genes were hypermethylated, 162 genes were hypomethylated, and 14 genes exhibited both hypermethylation and hypomethylation. The m6A was one of the important RNA modifications that affect RNA stability (half-life) [46]. In 162 hypomethylated genes, changes in the m6A levels of many genes have been found to affect their mRNA stability, such as *APC* gene (Table A3). Wang et al. found that METTL3 accelerated the degradation of *APC* mRNA in a YTHDF-dependent manner by up-regulating the m6A modification level of *APC* mRNA, thereby leading to oesophageal squamous cell carcinoma cell proliferation and tumour formation in mice [47]. Moreover, changes in the m6A levels of many of the above-hypermethylated genes have also been reported to affect their mRNA stability, such as *SREBF1* gene (Table A3). Tang et al. demonstrated that upregulation of FTO levels might promote liver steatosis by increasing the stability of *SREBF1* mRNA via demethylating its m6A sites [48]. However, whether the m6A methylation changes of these genes are related to the development of ARHL still needs further research in the future.

Among 262 genes with differential expression and differential m6A methylation simultaneously, some functional genes may be potential targets for future therapies, such as *EGFR* gene (Table A3). EGFR, a member of the epidermal growth factor receptor (HER) family, plays a key role in the regulation of cell proliferation and differentiation [49]. Previous studies have provided evidence that EGFR was required for the regenerative proliferation of auditory supporting cells in mammals and birds [50], and impaired glycolysis promotes alcohol exposure-induced apoptosis in HEI-OC1 cells by inhibiting EGFR signalling [51]. Interestingly, recent research has found that reader proteins IGF2BP3 or YTHDF2 also can directly bind to the m6A modification site of *EGFR* mRNA to change its stability [52,53]. However, the m6 A modification of *EGFR* mRNA in ARHL is still poorly understood and its role in ARHL is the next step in our research plan. High-throughput sequencing platforms may produce sequencing errors originating from the sequencing technology itself [54]. We validated differentially expressed genes in the combined MeRIP-Seq and mRNA-Seq analyses by MeRIP-qPCR and qRT-PCR, respectively, the results of which were consistent with those of high-throughput sequencing, further indicating the reliability of high-throughput sequencing in this study.

## 5. Conclusions

This study confirms that changes in m6A modification are strongly associated with ARHL and is the first to identify an m6A transcriptome pattern in ARHL mice. This study reveals the potential relationship between m6A methylation and mRNA expression in ARHL and provides new insights for further study of the pathogenesis of ARHL and potential therapeutic strategies.

## Figures and Tables

**Figure 1 biomolecules-13-01537-f001:**
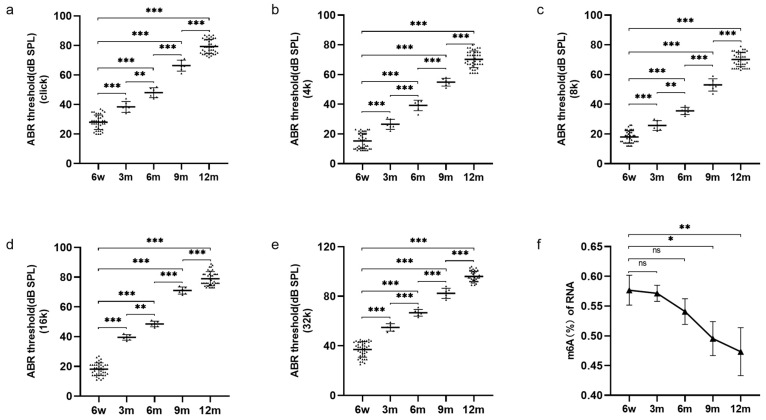
The change of ABR hearing thresholds and total m6A levels in C57BL/6J mice. (**a**–**e**) ABR thresholds were observed in 6 w (N = 39), 3 m (N = 6), 6 m (N = 6), 9 m (N = 6), and 12 m (N = 39) C57BL/6J mice at click, 4, 8, 16, and 32 kHz. (**f**) Quantification of m6A in total cochlear RNA was determined by colorimetric method in 6 w, 3 m, 6 m, 9 m, and 12 m C57BL/6J mice (N = 3). Abbreviations: w, week-old; m, month-old; ABR, Auditory brainstem response; N, the number of experiments. Notes: *p* < 0.05 represents a statistically significant difference; ns represents no statistically significant difference. * *p* < 0.05; ** *p* < 0.01; *** *p* < 0.001.

**Figure 2 biomolecules-13-01537-f002:**
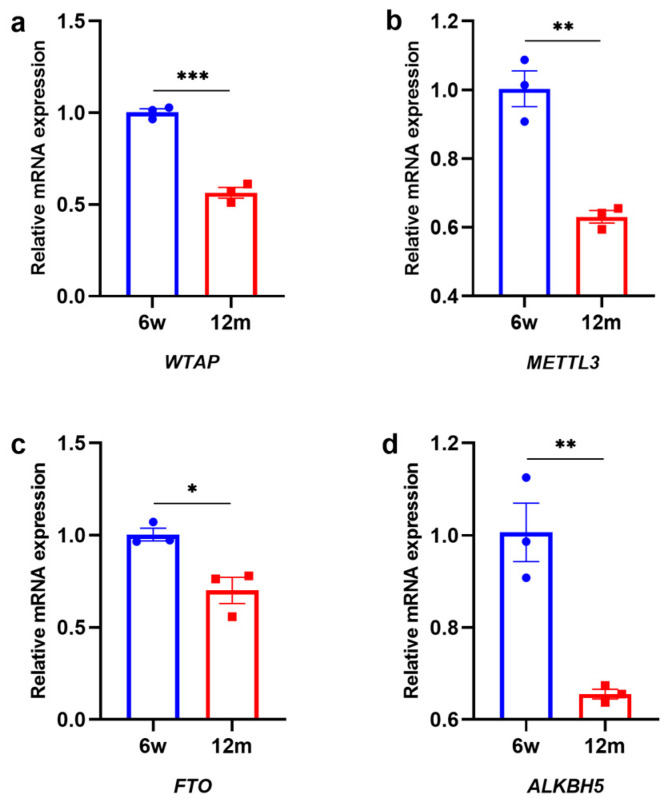
The RNA expression of m6A enzymes in 6 w and 12 m C57BL/6J mice. (**a**) *WTAP*, (**b**) *METTL3*, (**c**) *FTO*, and (**d**) *ALKBH5* in 6 w C57BL/6J mice compared with 12 m C57BL/6J mice (N = 3). Abbreviations: w, week-old; m, month-old; N, the number of experiments. Notes: *p* < 0.05 represents a statistically significant difference. * *p* < 0.05; ** *p* < 0.01; *** *p* < 0.001.

**Figure 3 biomolecules-13-01537-f003:**
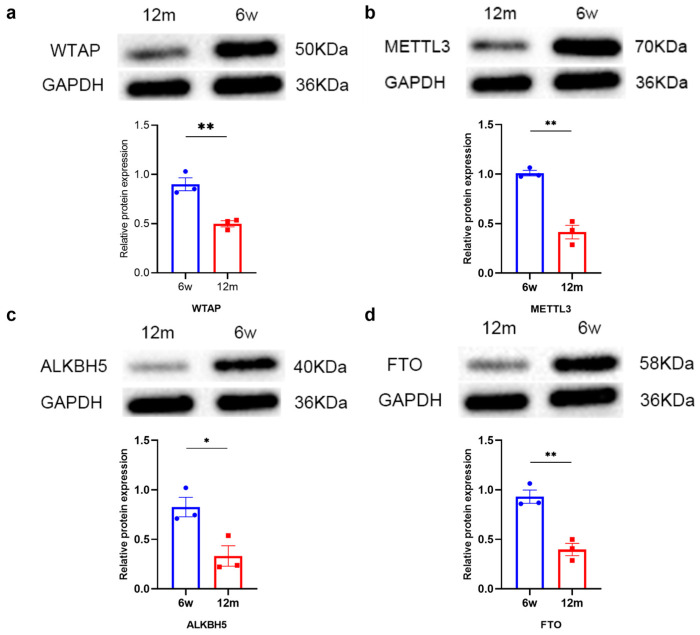
The protein expression of m6A enzymes in 6 w and 12 m C57BL/6J mice. (**a**) WTAP, (**b**) METTL3, (**c**) FTO, and (**d**) ALKBH5 in 6 w C57BL/6J mice compared with 12 m C57BL/6J mice (N = 3). Abbreviations: w, week-old; m, month-old; N, the number of experiments. Notes: *p* < 0.05 represents a statistically significant difference. * *p* < 0.05; ** *p* < 0.01. Western Blot images can be found in Appendix A.

**Figure 4 biomolecules-13-01537-f004:**
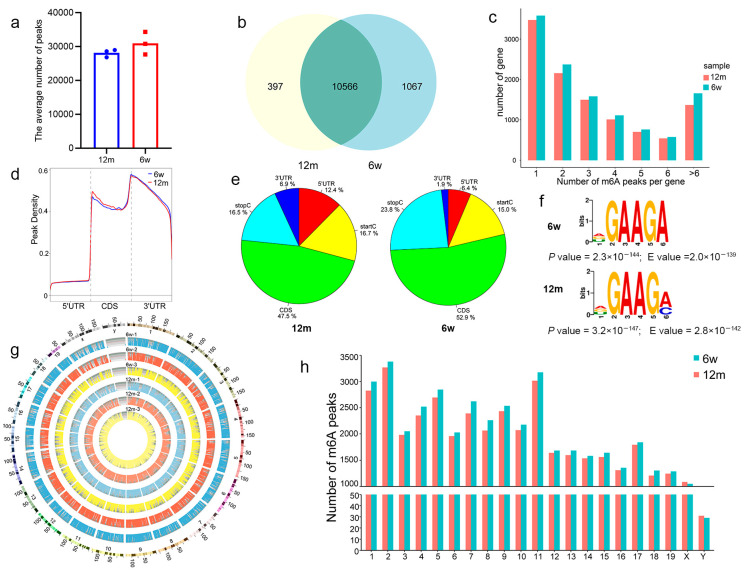
Overview of m6A methylation patterns in 6 w mice and 12 m mice. (**a**) The average number of m6A peaks in the 6 w mice and the 12 m mice. (**b**) Venn diagram of m6A-modified genes in the 6 w mice and the 12 m mice. (**c**) Number of peaks per gene. (**d**) Density of differential m6A peaks along transcripts. Each transcript was divided into five parts: 5′UTR, CDS, 3′UTR, startC, and stopC. (**e**) Pie charts showing the region of m6A peaks in each group. (**f**) The most conserved sequence motif of the differential m6A peak region. (**g**) The distribution patterns of m6A peaks in different chromosomes. (**h**) The count of m6A peaks in per chromosome. Abbreviations: w, week-old; m, month-old.

**Figure 5 biomolecules-13-01537-f005:**
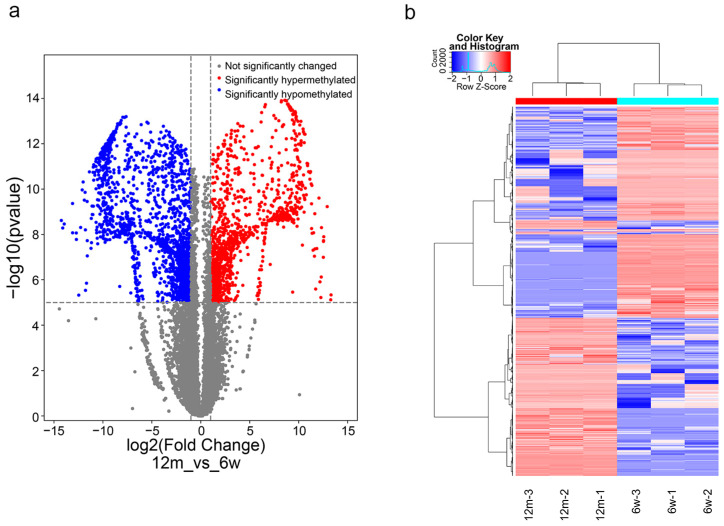
(**a**) Volcano plot representation of microarray data on the differentially expressed m6A methylation genes. (**b**) Hierarchical cluster analysis of differentially m6A methylation genes.

**Figure 6 biomolecules-13-01537-f006:**
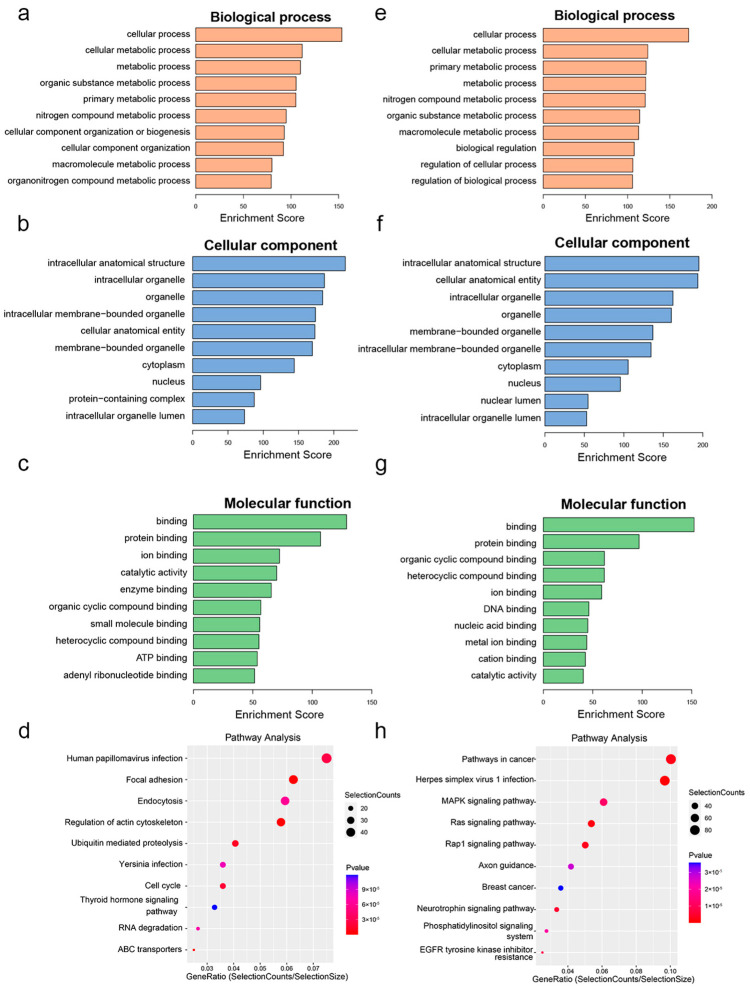
Gene ontology enrichment and pathway analysis of differentially methylated genes. (**a**–**c**) The top 10 GO terms significantly enriched for hypermethylated genes. (**e**–**g**) The top 10 GO terms significantly enriched for hypomethylated genes. (**d**) The top 10 KEGG pathways significantly enriched for hypermethylated genes. (**h**) The top 10 KEGG pathways significantly enriched for hypomethylated genes.

**Figure 7 biomolecules-13-01537-f007:**
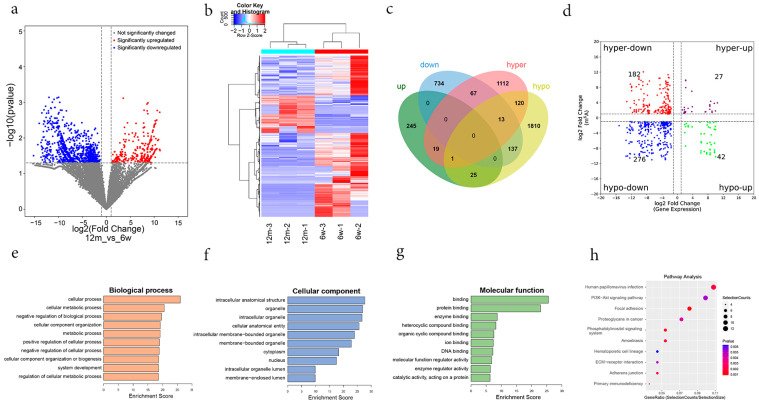
Conjoint analysis of m6A methylation and mRNA expression. (**a**) The volcano plot showing the differentially expressed mRNA gene. (**b**) Hierarchical cluster analysis of differentially expressed mRNA genes. (**c**) Venn diagram of genes with differentially m6A methylation and differentially expressed mRNA. (**d**) Four quadrant graphs of peaks with differentially m6A methylation and differentially expressed mRNA. (**e**–**g**) The top 10 GO terms significantly enriched for genes with differentially m6A methylation and differentially expressed mRNA. (**h**) The top 10 KEGG pathways significantly enriched for genes with differentially m6A methylation and differentially expressed mRNA. Notes: the purple color circles represent the upregulated genes with hypermethylation, the red color circles represent the downregulated genes with hypermethylation, the blue color circles represent the downregulated genes with hypomethylation, the green color circles represent the upregulated genes with hypomethylation.

**Figure 8 biomolecules-13-01537-f008:**
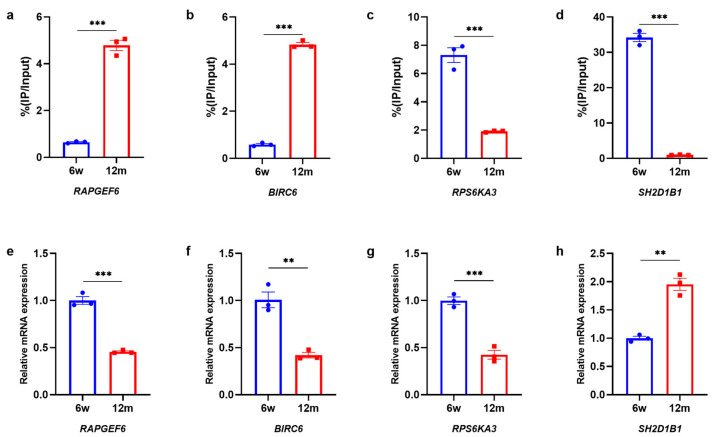
Verification of differentially m6A methylation level and differentially mRNA expression. (**a**–**d**) MeRIP-qPCR showing m6A enrichment in mRNA of *RAPGEF6*, *BIRC6*, *RPS6KA3*, and *SH2D1B1* genes (N = 3). (**e**–**h**) qRT-PCR showing mRNA expression of *RAPGEF6*, *BIRC6*, *RPS6KA3*, and *SH2D1B1* genes (N = 3). Abbreviations: N, the number of experiments. Notes: *p* < 0.05 represents a statistically significant difference.** *p* < 0.01; *** *p* < 0.001.

**Table 1 biomolecules-13-01537-t001:** The top 10 hypermethylated and hypomethylated genes among the 262 genes.

ID	Gene	m6A Fold Change	m6A *p*-Value	Gene Log FC	Gene *p*-Value	Change
ENSMUSG00000037533	*RAPGEF6*	433.2	1.2812 × 10^−14^	−10.025374	0.00213148	Hyper-down
ENSMUSG00000024073	*BIRC6*	800.7	5.0835 × 10^−14^	−2.9943806	0.02636128	Hyper-down
ENSMUSG00000058997	*VWA8*	201.1	7.6235 × 10^−14^	−3.7058447	0.01945395	Hyper-down
ENSMUSG00000068036	*MLLT4*	353.6	1.4383 × 10^−13^	−1.9414922	0.04990249	Hyper-down
ENSMUSG00000022607	*PTK2*	19.9	1.6866 × 10^−13^	−3.4652058	0.01968117	Hyper-down
ENSMUSG00000032410	*XRN1*	1434.8	2.9063 × 10^−13^	−10.023874	0.00555607	Hyper-down
ENSMUSG00000003847	*NFAT5*	8.6	3.6304 × 10^−13^	−2.3516677	0.03805174	Hyper-down
ENSMUSG00000032555	*TOPBP1*	11.8	8.4203 × 10^−13^	−7.2946474	0.01763448	Hyper-down
ENSMUSG00000024672	*MS4A7*	6.7	2.8421 × 10^−12^	1.83305783	0.04776993	Hyper- up
ENSMUSG00000029186	*PI4K2B*	3.5	2.8892 × 10^−12^	−10.125856	0.02227473	Hyper-down
ENSMUSG00000067149	*IGJ*	312.9	1.4276 × 10^−13^	8.49015602	0.00112192	Hypo-up
ENSMUSG00000031309	*RPS6KA3*	24.95	1.8979 × 10^−13^	−5.2427937	0.00759645	Hypo-down
ENSMUSG00000096334	*SH2D1B1*	412.2	3.1109 × 10^−13^	6.18769686	0.02234852	Hypo-up
ENSMUSG00000030231	*PLEKHA5*	167.2	3.5733 × 10^−13^	−6.9262512	0.00946439	Hypo-down
ENSMUSG00000024900	*CPT1A*	524.4	3.6441 × 10^−13^	−2.580643	0.01844615	Hypo-down
ENSMUSG00000025278	*FLNB*	93.24	3.6648 × 10^−13^	−3.7084202	0.03199615	Hypo-down
ENSMUSG00000030811	*FBXL19*	470.7	3.9417 × 10^−13^	−9.4987969	0.03882934	Hypo-down
ENSMUSG00000024749	*TMC1*	543.7	5.0403 × 10^−13^	−8.4734806	0.03470489	Hypo-down
ENSMUSG00000061080	*LSAMP*	554.1	5.7905 × 10^−13^	9.7209504	0.00629816	Hypo-up
ENSMUSG00000022297	*FZD6*	698.6	6.9979 × 10^−13^	−9.89231	0.0045972	Hypo-down

## Data Availability

All of the data generated or analyzed during this study are available from the corresponding author upon reasonable request.

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
