# Peer review of "Comprehensive Transcriptomic Profiling of m6A Modification in Age-Related Hearing Loss"

_biomolecules, 2023, doi:10.3390/biom13101537_

Round 1
Reviewer 1 Report
The manuscript by Feng et al. describes that the onset of age-related hearing loss (presbycusis) in C57Bl6 mice is paralleled by a reduction in overall N6-adenosine methylation (m6A) levels in inner ear RNAs. Further, the authors compared the transcriptomes of young (6-week-old) normal hearing mice with old (12 month) deaf mice (hearing thresholds over 80 db across all frequencies) and observed changes in specific m6A methylation patterns and also in expression levels of transcripts.
MAJOR CONCERN
Overall, the study is well designed and well executed, though the major problem is that the authors' main conclusion is "we observed changes" and they do not dwell on which changes are significant and why. This is very frustrating to the reader. For instance, a long paragraph is devoted to discuss the results of the genome wide m6A patterns and the differential expression RNAseq data separately (I wonder what is the interest there, in my humble opiniuon, that could be erased without any significant change). However, when they identify a set of 262 genes with significant differences in both m6A patterns and transcript expression, they fail to go into any details at all... Given that the addition of m6A modification is generally agreed to reduce the half-life of mRNA, it is essential that they describe which genes become hypo- or hypermethylated and whether this correlates with the observed fall or increase in expression levels. And, moreover, which cellular pathways are affected, how were they affected and what is the significance of the changes for age-related hearing loss.
MINOR CONCERNS
All throughout the manuscript and tables. Please italicize all gene names, so that we can easily follow whether you are talking about the gene or the protein.
All throughout Figure legends: please state how many experiments (N) were performed to obtain such mean results.
Line 41: "internal" should read "inner".
References 7, 8 and 9 are good. However, the negeral reader might benefit if the authors refer a recent review describing the roles of m6A methylation.
Line 69 "mettl3" should appear in capitals.
Line 81: Specifiy that the animal protocol was revised by an animal welfare committee.
Table 1 should be relocated to the end of the results section (page 12).
Line 132: please describe how was RNA fragmentation achieved. Otherwise, the protocol cannot be replicated.
Line 174: "upwards" should read "upwards shift".
Line 182-183: coincidence does not always mean causality. Please tone down this sentence.
Figure 3: How did you quantitate the Western blot results? This muts be described either here or in the methods section. In addtion, please label which bands belong to 6wk and which bands belong to 12m mice.
Lines 190-191. Please make an effort to distinguish genes from proteins to enable an easier understanding of your data.
Figure 6 and 7. lettering is so small that it is very hard to read. Please increase font size.
Figure 8. Sections f and g. The relative level of the control should be at 1.0. But the bar is higher. How come? Please check.
Line 341. The sentence should read "We found that the transcripts of 11666 genes had m6A modifications in 6 w mice and the transcripts of 10963 genes..."
The manuscript is quite well written. Yet, there are some inconsistencies with missing words that could be easily revised and solved.
Reviewer 2 Report
The research conducted by Feng et al. discusses the potential roles of m6A modifications in the context of age-related hearing loss (ARHL). Using the cochlea of the C57BL/6J mouse, a model characterized by early-onset hearing loss, the authors offer insights into the interplay between m6A and ARHL. Initially, they observe a decline in global m6A levels, implying m6A's involvement in ARHL development. Employing m6A-seq and RNA-seq techniques, the study showed how m6A could regulate gene expression in ARHL. They found differentially methylated genes that are also differentially expressed, particularly enriching for the PI3K-AKT signaling pathway. This investigation brings new insight into our understanding of m6A's role in ARHL. However, the authors could enhance their work by providing more comprehensive bioinformatic analyses and addressing the significance of genes regulated by m6A.
1. In the introduction, it would be beneficial for the authors to elaborate on the cochlea's importance in age-related hearing loss and elucidate their rationale for selecting it as the point of analysis in this study.
2. Despite METTL14 being a significant m6A regulator (line 52), its absence from the experiment investigating m6A modification enzyme and protein level changes warrants explanation, given its significance.
3. While C57BL/6J mice is a widely employed ARHL model and the cochlea is specifically targeted, the authors should consider potential confounding factors beyond age-related hearing loss, such as general aging effects. Clarification on isolating the effects of age-related hearing loss from aging itself would be valuable.
4. The study should detail the specific m6A antibody used. It is essential to indicate whether the same antibody was used for m6A-seq and m6A-qPCR, given the antibody's potential variability.
5. Precise and replicable methodology descriptions are essential, particularly for m6A-seq, m6A-qPCR, and bioinformatics pipelines. Parameters such as RNA input quantity for m6A-RIP should be provided to ensure accurate reproduction.
6. In Figure 7f, the authors illustrate m6A motif enrichment. While acknowledging the potential for the GAAG motif (line 352), further exploration is warranted. Parameters used in MACS peak calling, the impact of FDR cutoff for peak calling on motif enrichment, and the p-value score of the detected motif should be detailed for transparency.
7. N=6 was used for m6A-seq. How was a consensus peak set built to conduct differential peak analysis?
8. The authors performed m6A-qPCR and qPCR on the top hypermethylated and hypomethylated genes, respectively (line 296). Expanding on the functions of these genes and their implications in ARHL would provide context for the study's findings.
9. While indicating that m6A deposition in coding regions tends to stabilize mRNA (line 347), it is essential to acknowledge the dynamic nature of m6A at CDS regions, as supported by other literature.
10. Representing bar plots as bar-dot plots could enhance visualization consistency and accuracy throughout the manuscript.
Round 2
Reviewer 1 Report
I have read carefully the revised version of the manuscript. Now I have only two minor concerns:
(1) A broken sentence (no ending) at lines 360-361.
(2) Fonts are still too small to be read easily in some figures.
It is OK. Maybe a copyeditor should check for minor inconsistencies in font or abbreviations.
Reviewer 2 Report
The authors nicely reflected the suggestions. We have minor comments as below:
1. For the m6A-RIP experiments, it would be helpful to specify the exact amounts of m6A antibody and RNA input used. This information is vital as it can provide insights into the binding specificity and the reliability of the detected peaks.
2. The prominence of the GAAG motif as the top motif is somewhat surprising, given there aren’t many studies findings this motif, as mentioned in Comment 6. It's worth considering whether the less stringent nominal-pvalue cutoff of 0.00001 used in MACS peak calling might have contributed to this motif's prominence. Also, given the small number in replicate (n=3), further exploration of motif enrichment with different cutoffs could provide clarity.
3. In Figure 4A, depicting the variability of the number of peaks in each subject would enhance the presentation of the data, providing a clearer understanding of the sample-to-sample variation.
4. It would be beneficial if the authors could explain the significance of the number of m6A peaks on each chromosome in this study. Are there existing studies or literature that discuss the importance of m6A peaks in different chromosomes or is this an aspect for quality check.
